# Intercepted Photosynthetically Active Radiation (PAR) and Spatial and Temporal Distribution of Transmitted PAR under High-Density and Super High-Density Olive Orchards

**Adolfo Rosati** [1,*], **Damiano Marchionni** [1], **Dario Mantovani** [1], **Luigi Ponti** [2,3] and **Franco Famiani** [4]

1. Council for Agricultural Research and Economics (CREA), Research Centre for Olive, Fruit and Citrus Crops, via Nursina 2, 06049 Spoleto, Italy; d.marchionni@outlook.it (D.M.); mantdar2@gmail.com (D.M.)
2. Agenzia Nazionale per le Nuove Tecnologie, L'energia e lo Sviluppo Economico Sostenibile (ENEA), Centro Ricerche Casaccia, via Anguillarese 301, 00123 Rome, Italy; luigi.ponti@enea.it
3. Center for the Analysis of Sustainable Agricultural Systems (CASAS Global), 37 Arlington Avenue, Kensington, CA 94707, USA
4. Dipartimento di Scienze Agrarie, Alimentari e Ambientali, Università degli Studi di Perugia, 06121 Perugia, Italy; franco.famiani@unipg.it
* Correspondence: adolfo.rosati@crea.gov.it; Tel.: +39-0743-49743

**Abstract:** We quantified the photosynthetically active radiation (PAR) interception in a high-density (HD) and a super high-density (SHD) or hedgerow olive system, by measuring the PAR transmitted under the canopy along transects at increasing distance from the tree rows. Transmitted PAR was measured every minute, then cumulated over the day and the season. The frequencies of the different PAR levels occurring during the day were calculated. SHD intercepted significantly but slightly less overall PAR than HD ($0.57 \pm 0.002$ vs. $0.62 \pm 0.03$ of the PAR incident above the canopy) but had a much greater spatial variability of transmitted PAR (0.21 under the tree row, up to 0.59 in the alley center), compared to HD (range: 0.34–0.43). This corresponded to greater variability in the frequencies of daily PAR values, with the more shaded positions receiving greater frequencies of low PAR values. The much lower PAR level under the tree row in SHD, compared to any position in HD, implies greater self-shading in lower-canopy layers, despite similar overall interception. Therefore, knowing overall PAR interception does not allow an understanding of differences in PAR distribution on the ground and within the canopy and their possible effects on canopy radiation use efficiency (RUE) and performance, between different architectural systems.

**Keywords:** grove; light; radiation use efficiency; olea europaea

## 1. Introduction

The photosynthetically active radiation (PAR) intercepted by a crop canopy is one of the main factors determining biomass production, being the source of energy for the process of photosynthesis [1]. Linearity between biomass and cumulated intercepted radiation has been found both for herbaceous crops (e.g., beans [2]; soybean [3]; lettuce [4]) and for tree species, particularly in apple [5,6], but also in olive [7,8]). In fact, in the absence of other limitations, like water, nutrients or temperature limitations, the crop dry matter production can be modeled as a function of the cumulated PAR incident on the crop, the fraction of this that is intercepted by the canopy and its use efficiency (i.e., radiation use efficiency: RUE) [9], where RUE is the ratio of the dry matter produced per unit of radiation intercepted. Therefore, measuring PAR interception is important for the estimation of potential growth and yield of crops (crop modeling) for both herbaceous and tree crops. The fraction of PAR that is intercepted by the crop depends on the characteristics of the incident radiation, but also, and above all, on the optical and architectural properties of the tree [10]. Unlike most herbaceous crops, tree crops, including olive orchards, have discontinuous canopies, making the assessment of PAR interception and transmission difficult to both

model and measure [5,11–13]. Probably due to these difficulties, only few authors have worked on either modeling or measuring PAR interception in different olive orchards, ranging from extensive systems to high-density ones. For instance, Mariscal et al. [11], reported fractions of PAR transmittance ranging between about 0.05–0.85 (i.e., interception of 0.95–0.15, respectively) in orchards ranging from 0.1–0.7 of canopy ground cover. These values were used to validate a model of PAR interception in olive groves. Using this model, Villalobos et al. [8], estimated PAR interception in a variety of olive groves, from very extensive (spacing: 12 by 12 m) to high density (6 by 3.7 m), finding values ranging from about 0.20–0.65. In the last few decades, a new typology of olive orchard has been developed: the super high-density (SHD) orchard [14,15]. These orchards are characterized by much higher tree densities than in traditional or even high-density (HD) olive orchards, up to 1500–2200 trees per hectare [14]. In SHD systems the tree canopies form a continuous hedge of reduced size, which allows continuous mechanical harvesting with over-the-row straddle machines. This creates canopies that are discontinuous across tree rows, but continuous along the row, providing for a partial simplification of irradiance modeling. One objective when designing such systems, in olive [16] as well as in apple [5,6] and other species, is to ensure adequate canopy light interception by the entire canopy. In fact, while the increase in tree density leads to greater light interception and increased productivity in the early stages of the orchard [17], mutual (and self) shading increases as the plants grow [17,18]. Too much shade can depress both flowering and fruiting processes in olive [19–21] and a drastic reduction in PAR (i.e., below 10% of the incident PAR) increases olive leaf senescence [22]. Even when fruiting occurs, yield and oil concentration decrease with irradiance below 40–60% of incident PAR [18,23–25]. Similar findings have long been reported in other species, such as apple, where yield correlates well with canopy light interception up to medium interception values (50–60%) while at greater canopy densities this correlation may fail, and better correlations are found between yield and light interception by spur leaves (5–6). That is because part of the light is intercepted by elongation shoot leaves, which usually do not contribute carbon to fruits. Olives do not have spurs, yet the light intercepted by parts of the canopy that are too shaded will not contribute to fruit production. Therefore, it is important that the architectural design of SHD olive orchards allows irradiance levels above certain thresholds also at the base of the canopy [26]. This can be achieved by designing the orchard with adequate canopy height, slope, width and alley width [14,26].

Several studies have assessed, by measuring and/or modeling, irradiance levels in different positions within the canopy or on the canopy faces, with the aim of correlating irradiance levels to fruit quantity and quality parameters [24,25,27,28] or to morphological and physiological parameters [29]. Surprisingly, however, data on the overall PAR interception of SHD olive orchards are very scarce and have been obtained only by modeling (e.g., [24]). In addition, when comparing PAR interception between different orchard systems, like traditional systems (where the tree canopy is discontinuous in both directions) and SHD systems (forming continuous hedgerows), knowing the overall PAR interception may provide only partial understanding of PAR distribution. In fact, for a given overall PAR interception, PAR distribution within the canopy and on the ground is likely to differ, due to the very different architectural characteristics. No data are available on the spatial and temporal patterns of PAR transmission under the canopy of the different olive systems.

In this study we compared PAR interception in a high-density (HD) and a super high-density (SHD) olive system during the period March–September.

The specific objectives were to asses:

- the PAR intercepted by the two systems (HD and SHD);
- the spatial distribution of transmitted PAR on the ground;
- the temporal variation during the day (class frequency distribution) of the transmitted PAR.

## 2. Materials and Methods

### 2.1. Experimental Site

The experiment was carried out at Colle Cecco, near Spoleto, central Italy (42°48′34.4″ N 12°39′39.7″ E). The site is characterized by a mean yearly precipitation of 816 mm, with daily mean minimum and maximum air temperature of $7.4 \pm 5.4$ and $18.8 \pm 7.7$ °C, respectively (data from 1985–2014). The soil is a calcareous marl/marlstone, more than 120 cm deep, with bulk density of 1.5 and 1.6 g cm$^{-3}$ when measured at 20 and 40 cm depth respectively. The aspect is south-south-west (SSW), with a 10% slope. We investigated a high-density (HD) and a super high-density (SHD) olive orchard system. For each system, the orchard chosen is representative of the typical respective system, in terms of tree age, spacing, canopy size and architecture. In the SHD system, the 8-year-old trees were spaced 1.5 m along the row, and 4.0 m between rows, forming a continuous hedge 2.5 m high and 1.2 m wide. In the HD system, the 20-year-old trees were trained to a vase and spaced 3.5 m along the row and 5 m between the rows. Trees were regularly pruned to maintain canopy diameters of about 3.5–4 m, forming a continuous canopy along the row, with maximum height of about 5 m, and a small gap between rows of about 1 m. A schematic representation and pictures of the two systems are shown in Figure 1. For both treatments, measurements were carried out in the two alleys between 3 adjacent rows of trees. Both systems had identical row orientation: NNE–SSW.

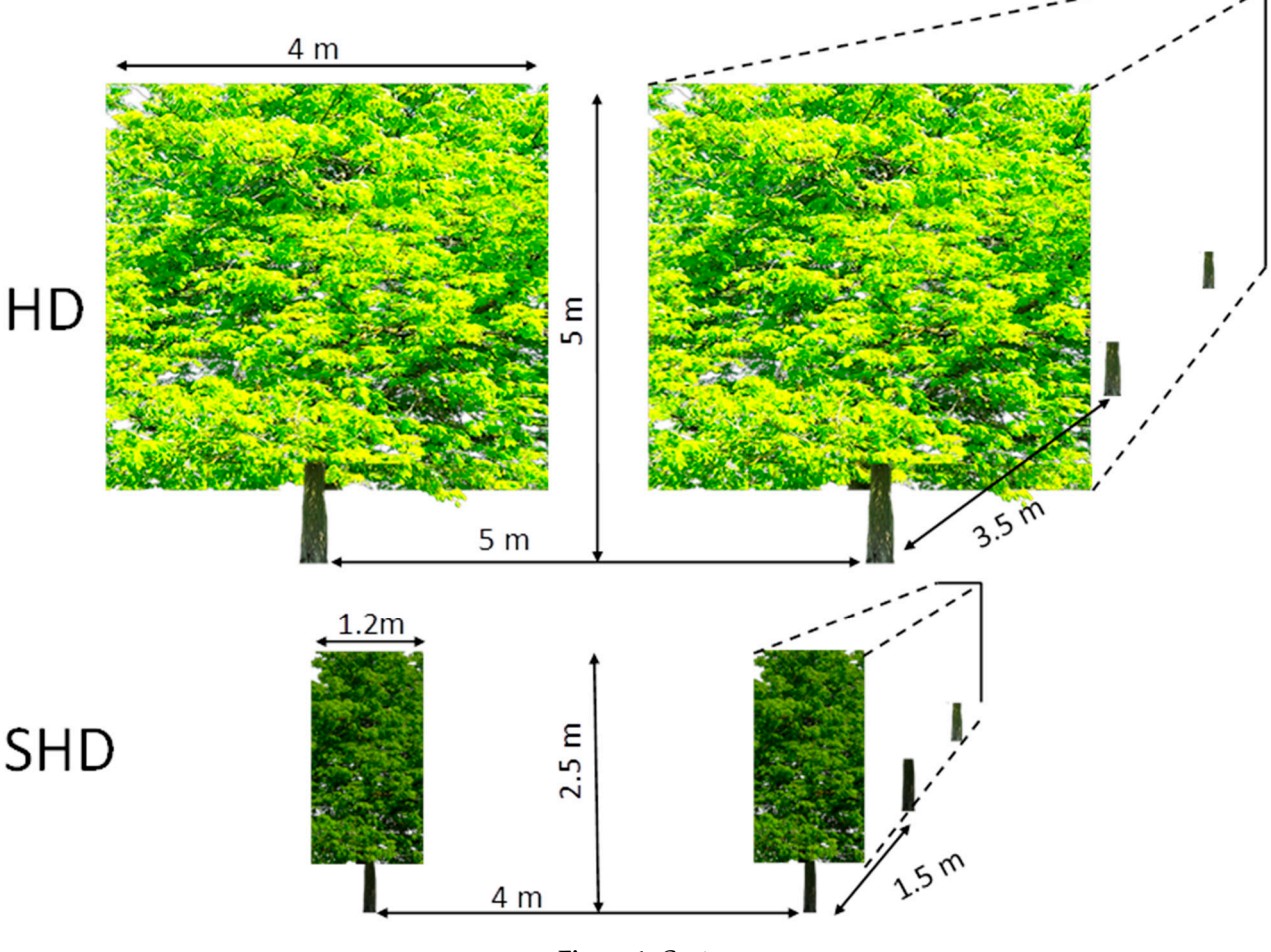

**Figure 1.** *Cont.*

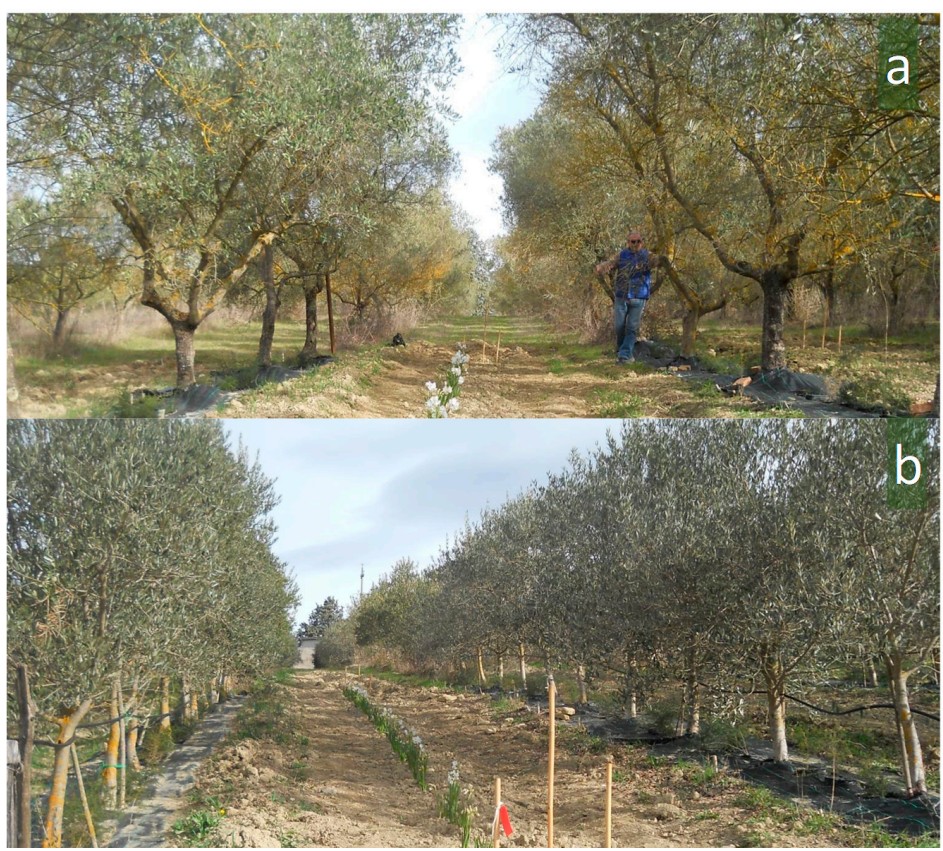

**Figure 1.** Top: Schematic representation of the two orchard systems studied: a high-density (HD) and a super high-density (SHD) system. Tree age and spacing and canopy height are representative of typical, mature, HD and SHD system, respectively. Bottom: pictures of the HD (**a**) and SHD (**b**) orchards.

*2.2. Incident and Transmitted Photosynthetically Active Radiation (PAR) Measurements*

In each orchard (i.e., HD and SHD) the PAR transmitted below the trees was measured along four transects (i.e., four replications) across tree rows, placing calibrated photosensors (GaAsP, Hamamatsu, Japan) horizontally on the ground below the tree row and at increasing distance on both the east and west side of the central tree row, up to the middle of the alleys. Sensors were placed 0.5 m apart; therefore, nine sensors were used in the 4 m transects in SHD and 11 sensors in the 5 m transects in the HD. As a convention, the distances from the tree row towards east and west were considered positive and negative, respectively. The photosensors were connected to a data logger (DL6, Delta-T, Cambridge, UK) taking records every minute from dawn to dusk. Measurements were taken from March to September during two days per month, in each of the two orchards. For each measuring day, one sensor was placed above the orchard to measure the PAR incident above the canopy.

For each position in the transects, the daily transmitted PAR was calculated by integrating the instantaneous measurements over the day. The absolute values were also expressed as a fraction of the daily PAR incident above the trees on the same day (from the above-canopy sensor). To estimate the monthly transmitted PAR (TPARmonthly) in each position along the transects, the overall PAR fraction for each month (i.e., average of the two measuring days) was multiplied by the monthly incident PAR, obtained from a nearby weather station located at the Papiano experimental station of the Department of Agricultural, Food and Environmental Sciences of the University of Perugia. The monthly values were summed up for the measuring season (March–September), obtaining the seasonal transmitted PAR (TPARseasonal). TPARseasonal for each position was also divided by the

seasonal incident PAR (from the weather station) to calculate the overall seasonal fraction of transmitted PAR in each position.

### 2.3. Class Frequency Distribution of Transmitted PAR Values

To further investigate the characteristics of the PAR transmitted under the two olive systems (HD and SHD) we analyzed the class frequency distribution of transmitted PAR values, by assessing, for each position along the transects, the daily frequency of PAR values (measured every minute) within classes of transmitted PAR.

The PAR classes considered were 0–50, 50–200, 200–400, 400–600, 600–800, 800–1000, 1000–1200, 1200–1400, 1400–1600, 1600–1800, >1800 $\mu$mol m$^{-2}$ s$^{-1}$. For the first class the value of 50 was chosen because it represents the leaf photosynthetic light compensation threshold for olive [30]: PAR levels in this class (i.e., 0–50 $\mu$mol m$^{-2}$ s$^{-1}$) result in negative net assimilation.

### 2.4. Statistical Analyses

Data are presented as means ± standard error. Comparison between SHD and HD data, and among positions in the transects within each system, were statistically analyzed by analysis of variance (ANOVA), according to a completely randomized design and the averages were compared by the Student-Newman-Keuls test.

## 3. Results

### 3.1. Spatial and Seasonal Patterns of Transmitted PAR

For the SHD system the position with the lowest cumulative transmitted PAR during the March–September period (TPARseasonal) was the one under the tree row, where TPARseasonal was 1850 mol m$^{-2}$ season$^{-1}$ (Figure 2a) when expressed in absolute terms, or 0.21 when expressed as a fraction of the incident PAR (Figure 2b). Moving towards the center of the alleys, TPARseasonal increased, reaching the highest value in the center of the alley on the west side (5365 mol m$^{-2}$ season$^{-1}$/0.62) and at 1, 1.5 and 2 m from the tree row in the east side (4772/0.55; 5164/0.60; 4837/0.56 respectively). Data show asymmetry between the two sides: the average TPARseasonal for all positions on the west side vs. those on the east side (excluding the position under the tree row) was 3704 mol m$^{-2}$/0.43 vs. 4512 mol m$^{-2}$/0.52 respectively. This is due to the tree row orientation which was NNE–SSW, exposing the east side of the rows, slightly oriented towards south, to greater irradiance than the west side, slightly oriented towards north, in agreement with model predictions [14,24,28]. Overall, TPARseasonal was very (and significantly, $p < 0.001$) variable in the SHD system, ranging from 1850 mole m$^{-2}$/0.21 under the trees to 5101/0.59 in the center of the alley (i.e., average of the positions in the center of the alley of both sides).

Contrary to the SHD system, in the HD system TPARseasonal was much more homogeneously distributed under the trees, with a narrow (and not significantly different) range of variation from 2913 mol m$^{-2}$/0.34 at −0.5 m from the tree rows, to 3685/0.43 in the center of the alley (i.e., average of the two alley centers).

The two orchards showed a similar seasonal pattern of the overall transmitted PAR (i.e., integrated over the whole transect: TPARmonthly) (Figure 2c). This pattern was similar to that of the incident PAR, with increasing trends in spring, then decreasing from July to September, in accordance with model predictions [14]. However, when expressed as a fraction of the incident PAR, TPARmonthly decreased over the season (Figure 2d). This was due to the fact that trees were pruned before the beginning of the experiment and the canopies grew during the season, decreasing the fraction of PAR transmitted. The overall transmitted PAR (i.e., integrated over the whole transect and cumulated for the whole season), was significantly ($p < 0.05$) but slightly lower for the HD olive orchard (3293.4 ± 231.8 mol m$^{-2}$/0.38 ± 0.03) than for the SHD (3701.7 ± 18.8 mol m$^{-2}$/0.43 ± 0.002).

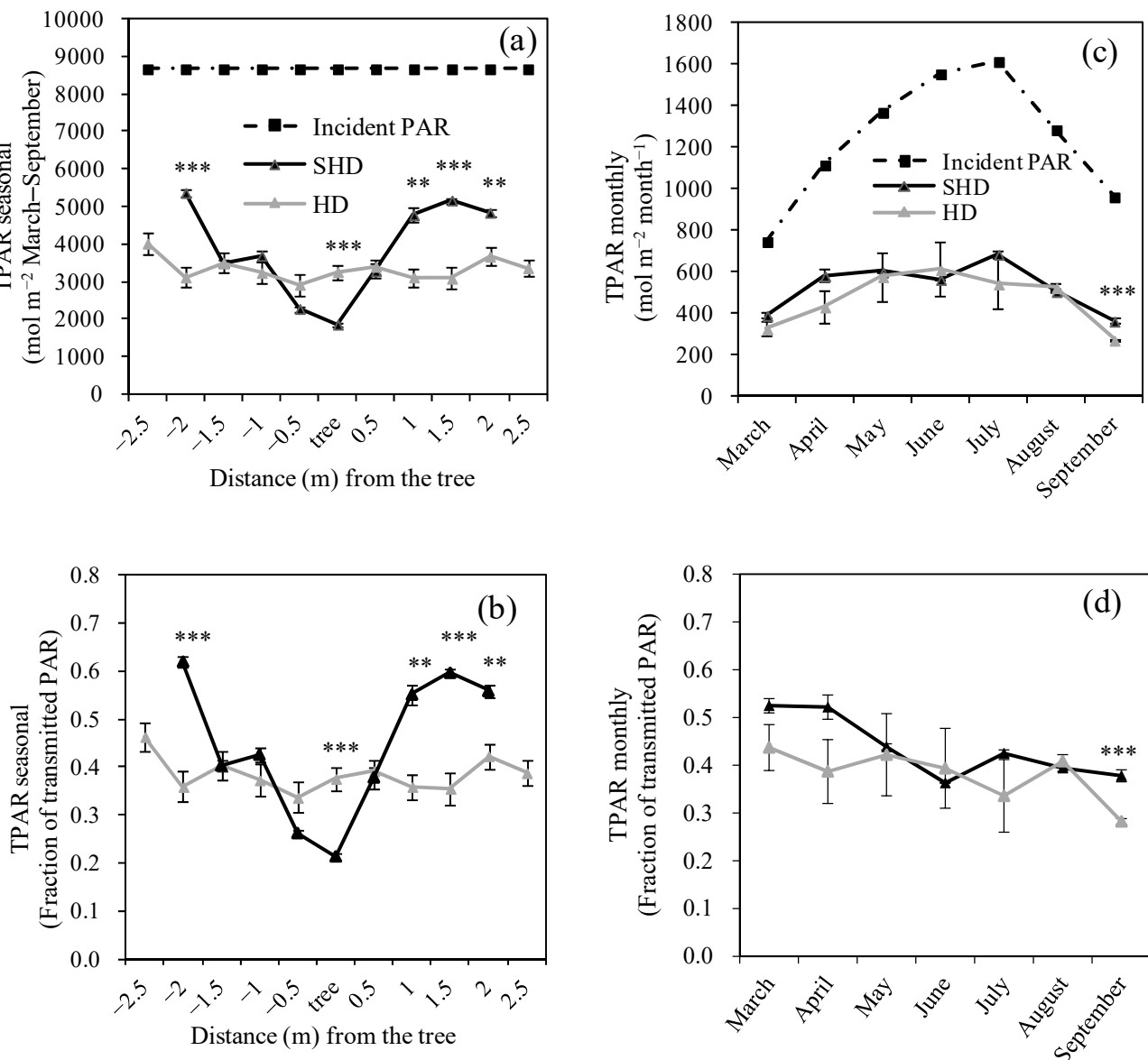

**Figure 2.** Seasonal (March–September) transmitted photosynthetically active radiation (TPARseasonal) in a high-density (HD) and a super high-density (SHD) systems at increasing distance (0.5 m steps) from the tree row, on both the east (negative distances) and west (positive distances) side, up to the middle of the alleys (i.e., at 2 and 2.5 m from the tree row, respectively for the SHD and the HD systems), expressed (**a**) in absolute terms and (**b**) relative to the photosynthetically active radiation (PAR) incident above the canopy obtained from a weather station. Monthly transmitted PAR (integrated over the different transect positions, TPARmonthly) expressed (**c**) in absolute terms and (**d**) relative to the PAR incident above the canopy. Values are means of 4 replications; bars indicate standard errors. Statistical significance between systems, within each position, was tested by analysis of variance (ANOVA, * = $p < 0.05$; ** = $p < 0.01$, *** = $p < 0.001$). For details on calculations of monthly transmitted PAR, see materials and methods, Section 2.2.

### 3.2. Class Frequency Distribution of Incident and Transmitted PAR Values

The class frequency distributions of the incident PAR values measured above the canopy differed substantially with season and between clear (sunny) and overcast (cloudy) days. To exemplify this, Figure 3 shows data from three sample days: two in March (one clear and one uniformly overcast) and one in May (clear). March was chosen as an example, because it was the month with the shortest (i.e., darkest) days in the dataset, and we also happen to have a clear day and a uniformly overcast day in the same month, so we could compare them. The clear day in May was chosen to compare a clear day in a period of long days (May) with a clear

day in a period of short (dark) days (March). We chose May instead of June, when days are longest, because we happened to not have perfectly clear days in June, so May had the longest perfectly clear day in the dataset. On the overcast day, the most frequent incident PAR values were those of relatively low intensity (200–400 $\mu$mol m$^{-2}$ s$^{-1}$). The frequencies decreased for both lower and higher PAR classes. This pattern is explained by the fact that clouds prevent the passage of direct beam radiation with high intensities and the light environment is characterized mainly by low-intensity diffuse radiation. By contrast, on a clear day in the same month (i.e., March), the most frequent values were those in the highest classes of the PAR range available that day (1200–1400 and 1400–1600 $\mu$mol m$^{-2}$ s$^{-1}$), with a relatively uniform distribution of PAR values in lower classes. Compared to the overcast day, the clear day had higher frequencies of PAR values in the classes above the 600–800 $\mu$mol m$^{-2}$ s$^{-1}$, and lower frequencies in the classes below. The clear day in May has a pattern similar to the clear day in March, but skewed towards higher classes because of the higher sun elevation.

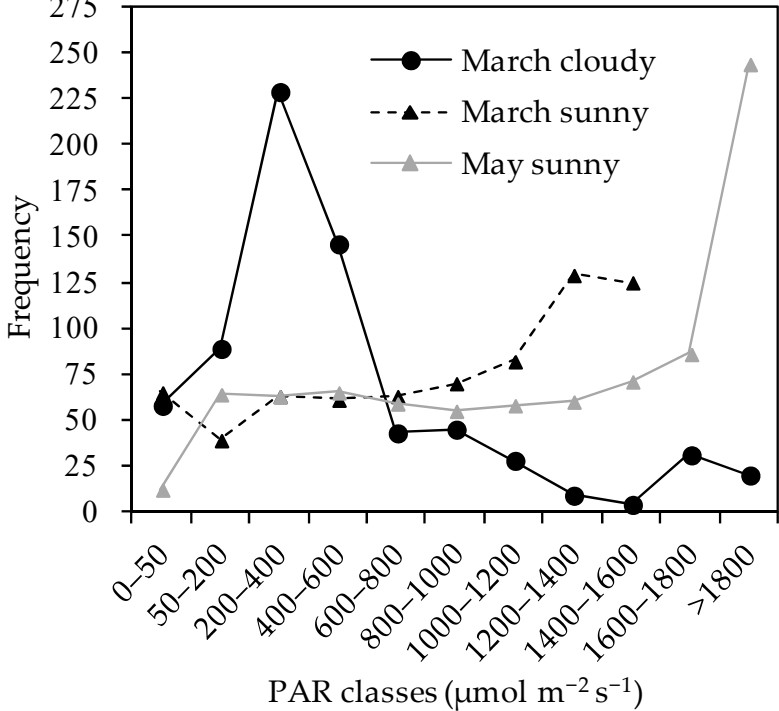

**Figure 3.** Class frequency distributions of incident PAR values measured above the canopy every minute from dawn to dusk, in three different days, two in March (one clear and one overcast) and one in May (clear). Each point is not an average, but is the frequency for the individual sample day, therefore standard error cannot be calculated. Reasons for showing these three individual sample days are explained in the text.

Figure 4 shows the mean class frequency distributions of transmitted PAR values for the different positions under the canopy, compared to the mean distribution for the incident PAR values on the same days, for both orchards. Given the large amount of data, for the sake of clarity only class frequency distribution data for the positions on the west side are shown; the results for the east side are very similar (data not shown). Data are means of the daily frequencies measured on all measuring days (i.e., two days per month and per treatment, for seven months). Given the different distributions between clear and overcast days, separate means were calculated for clear and overcast days. On overcast days all positions in the HD system experienced mostly low PAR values, reflecting the trend of the incident PAR, but skewed towards lower PAR classes (Figure 4a and Table 1). The most frequent PAR values were in the 200–400 class for the incident PAR and in the 50–200 for all positions in the transect. The situation is similar for the SHD systems (Figure 4b and

Table 1), but with more marked and mostly significant differences between the different positions: the most shaded positions (e.g., below the tree and at −0.5 m: black lines) were much more skewed towards lower classes, compared to the most illuminated positions (e.g., alley center, −1.5 m and −1 m: grey lines).

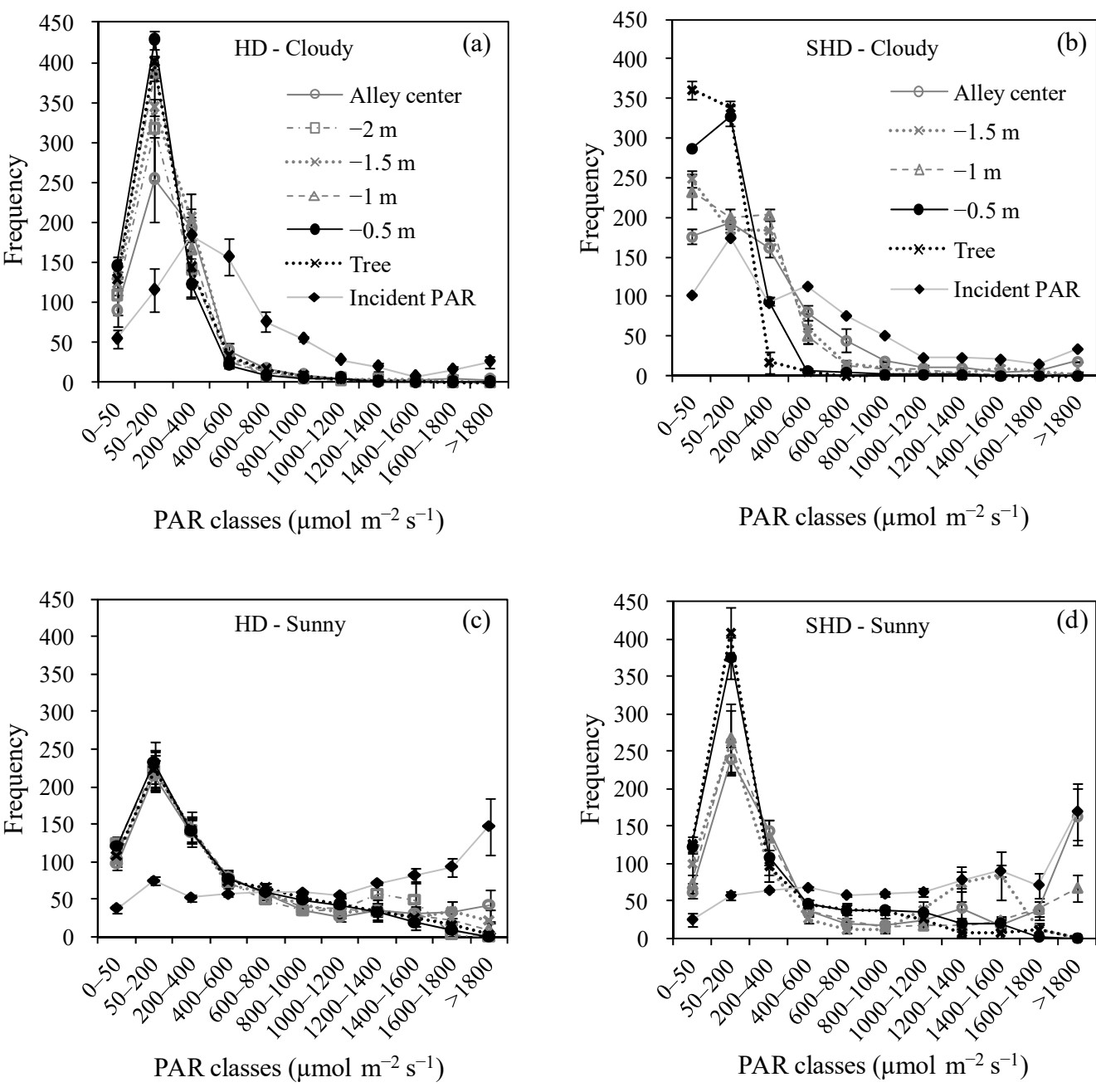

**Figure 4.** Class frequency distribution (mean of all available data) of incident (i.e., above the canopy) and transmitted PAR values as measured every minute, from dawn to dusk, at increasing distance (0.5 m steps) from the tree row along the west side of the transects, up to the middle of the alley, in a high-density (HD) and a super high-density (SHD) system. Data are means of all measurement days, separated in overcast (**a**,**b**) and clear or sunny (**c**,**d**) days, in both the HD (**a**,**c**) and the SHD (**b**,**d**) systems. Due to limitations in datalogger channels, measurements were made in different days for the two systems, as shown by the slightly different class frequency distribution patterns for the incident PAR. For statistical differences between positions, within each PAR class, see Table 1.

**Table 1.** Statistics for the position effect, within each PAR class, for the data shown in Figure 4. Statistical significance within each PAR class and within each grouping of data (i.e., HD Cloudy, SHD Cloudy, HD Sunny and SHD Sunny) was tested by analysis of variance (ANOVA, * = $p < 0.05$; ** = $p < 0.01$, *** = $p < 0.001$). When ANOVA gave significant effects, differences between positions were tested using the Student–Newman–Keuls test ($p = 0.05$). Different letter within −each PAR class and within each grouping of data indicate significant differences. For the SHD Cloudy data, no statistics are reported for incident PAR as the dataset contained only one day.

| Data Subset | PAR Class | | | | | | | | | | |
|---|---|---|---|---|---|---|---|---|---|---|---|
| HD Cloudy | 0–50 | 50–200 | 200–400 | 400–600 | 600–800 | 800–1000 | 1000–1200 | 1200–1400 | 1400–1600 | 1600–1800 | >1800 |
| *p* level | * | *** | ns | *** | *** | *** | *** | *** | ns | *** | *** |
| Alley center | ab | b | a | a | a | a | a | a | a | a | a |
| − 2 m | b | b | a | a | a | a | a | a | a | a | a |
| − 1.5 m | b | b | a | a | a | a | a | a | a | a | a |
| − 1 m | b | b | a | a | a | a | a | a | a | a | a |
| − 0.5 m | b | b | a | a | a | a | a | a | a | a | a |
| Tree | b | b | a | a | a | a | a | a | a | a | a |
| Incident PAR | a | a | a | b | b | b | b | b | a | b | b |
| | | | | | | | | | | | |
| SHD Cloudy | 0–50 | 50–200 | 200–400 | 400–600 | 600–800 | 800–1000 | 1000–1200 | 1200–1400 | 1400–1600 | 1600–1800 | >1800 |
| *p* level | ** | *** | *** | * | * | ** | ns | ** | * | ns | *** |
| Alley center | a | a | c | b | b | c | a | b | ab | a | c |
| − 1.5 m | b | a | c | ab | a | b | a | a | b | a | a |
| − 1 m | b | a | c | ab | a | ab | a | a | a | a | b |
| − 0.5 m | b | b | b | a | a | a | a | a | a | a | a |
| Tree | c | b | a | a | a | a | a | a | a | a | a |
| | | | | | | | | | | | |
| HD Sunny | 0–50 | 50–200 | 200–400 | 400–600 | 600–800 | 800–1000 | 1000–1200 | 1200–1400 | 1400–1600 | 1600–1800 | >1800 |
| *p* level | *** | * | * | ns | ns | ns | ns | ns | ns | *** | *** |
| Alley center | b | b | b | a | a | a | a | a | a | a | a |
| − 2 m | b | b | b | a | a | a | a | a | a | a | a |
| − 1.5 m | b | b | b | a | a | a | a | a | a | a | a |
| − 1 m | b | b | b | a | a | a | a | a | a | a | a |
| − 0.5 m | b | b | b | a | a | a | a | a | a | a | a |
| Tree | b | b | b | a | a | a | a | a | a | a | a |
| Incident PAR | a | a | a | a | a | a | a | a | a | b | b |
| | | | | | | | | | | | |
| SHD Cloudy | 0–50 | 50–200 | 200–400 | 400–600 | 600–800 | 800–1000 | 1000–1200 | 1200–1400 | 1400–1600 | 1600–1800 | >1800 |
| *p* level | *** | *** | ns | ** | ** | *** | ns | * | ** | ** | *** |
| Alley center | ab | b | a | a | ab | ab | ab | ab | ab | ab | b |
| − 1.5 m | b | bc | a | a | a | a | ab | ab | b | a | a |
| − 1 m | b | bc | a | a | ab | a | a | ab | ab | ab | a |
| − 0.5 m | b | c | a | ab | bc | bc | ab | ab | ab | a | a |
| Tree | b | c | a | ab | bc | bc | ab | a | a | a | a |
| Incident PAR | a | a | a | b | c | c | b | b | b | b | b |

On clear days (Figure 4c,d) both the incident PAR and the PAR transmitted in all positions are skewed toward higher PAR classes compared to the overcast days (Figure 4a,b), but again the transmitted PAR in the different positions is skewed more towards lower classes compared to the incident PAR. As for overcast days, in the HD system the patterns among the different positions are more uniform (and not significantly different), reflecting the more uniform PAR transmittance (Figure 2), while in the SHD system there are larger and mostly significant differences, with the most shaded positions more skewed towards mid and lower classes, compared to the most exposed positions.

To better understand how the different systems (HD and SHD) influence the characteristics of the transmitted PAR, we compared the frequency distributions of the transmitted PAR values for the two systems (clear days) only for the extreme positions: alley centers (−2.5 m from tree row in the HD and −2 m in the SHD system, Figure 5a) and below the tree (Figure 5b).

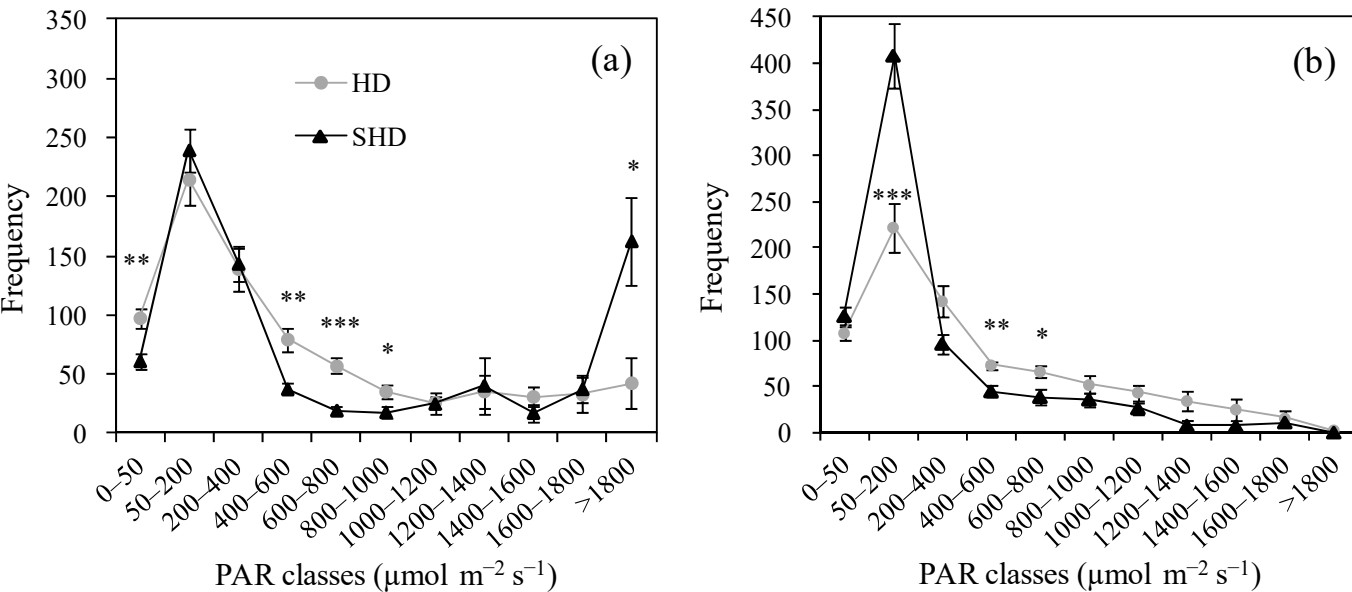

**Figure 5.** Class frequency distribution (mean of all available data) of transmitted PAR values as measured every minute, from dawn to dusk, as in Figure 4, but only for clear days, and for two positions along the transect, i.e., (**a**) the alley center (the position with the highest $TPAR_{seasonal}$, 2.5 m from tree row in HD system and at 2 m in SHD, and (**b**) the position below the tree. Statistical significance between systems, within each position, was tested by analysis of variance (ANOVA, * = $p < 0.05$; ** = $p < 0.01$, *** = $p < 0.001$).

The frequency distribution of the PAR transmitted in the alley center (Figure 5a) shows a bimodal pattern in the SHD system, with highest frequencies of low (50–200 µmol m$^{-2}$ s$^{-1}$) and high PAR values (>1800 µmol m$^{-2}$ s$^{-1}$). In the intermediate classes the frequencies are much lower. In the HD system, PAR is more uniformly distributed, with higher frequencies of intermediate PAR values (+107 min over the range 400 < PAR < 1600) and much lower frequencies (i.e., −120 min) in the highest PAR class (PAR > 1800). Therefore, in SHD, the most exposed positions (alley centers) tend to receive either very high PAR levels, or to be deeply shaded, while the situation is less extreme for the HD. This is due to the greater discontinuity of canopy cover in the SHD system, given by the narrow and dense canopy. In a tree row oriented roughly N–S, as in this experiment, the center of the alley is shaded when the sun is at low elevations (i.e., morning and evening), that is when incident PAR is low, making this position experience very low PAR at those times. At higher sun elevation, when incident PAR is high, this position is no longer shaded, thus receiving very high PAR. This explains the accentuated bimodal pattern of transmitted PAR in the alley center. The more uniform canopy cover of the HD system (i.e., the canopy gap between rows is about 1 m, compared to 2.8 m in the SHD, see Figure 1) reduces this phenomenon. Differences in canopy height also contributes to this: a higher canopy (5 m in the HD system) results in shadows that move faster and further, equalizing ground illumination, compared to shorter canopies (2.5 m in the SHD system), where shade concentrates more closely under the trees.

Compared to the alley centers (Figure 5a), the class frequency distribution of transmitted PAR under the trees is skewed towards lower PAR classes in the SHD system (Figure 5b). In fact, the accumulated frequencies for the lowest 4 classes totalize about 450 min for the alley centers (Figure 5a) vs. 680 min (over 4 extra hours) under the tree (Figure 5b). The top four classes totalize about 270 min for the alley center vs. less than 20 min (more than 4 h less). In the HD system, the position under the tree is less skewed towards lower PAR classes than the same position in SHD (Figure 5b), with much lower frequencies (−186 min in the 50–200 µmol m$^{-2}$ s$^{-1}$ class, and higher frequencies in the intermediate and high PAR classes (+184 min in the range 200 < PAR < 1800). This is due to the fact that the most shaded position in the SHD system (i.e., under the tree row) is almost always deeply shaded, including at mid-day when incident PAR is high, but the

direct radiation has to penetrate through the longest path within the narrow but relatively tall canopy. In the HD system, the more uniformly distributed and taller canopy results in more uniformly transmitted PAR on the orchard floor and thus less dense shade under the trees. Therefore, the greater shade under the tree row in the SHD system, compared to the HD system (Figure 2a,b), resulted not only in lower daily and seasonal PAR levels, but also in much higher frequencies of very low PAR (50–200 PAR) intensities, and lower frequencies of intermediate and high PAR levels, compared to the HD system (Figure 5b).

## 4. Discussion

### 4.1. Spatial distriButions of Transmitted PAR

The SHD system transmitted a significantly but slightly higher overall fraction (i.e., integrated over the transect) of seasonal PAR than the HD system, implying a slightly lower PAR interception (0.57 vs. 0.62). The value of PAR interception for the HD system (0.62) is well within the range of measured values reported by Mariscal et al. [11] and close to the upper range of modeled values reported by Villalobos et al. [8]. This is not surprising considering that tree density in our HD orchard was close and indeed greater than the upper values in the range of scenarios considered by Villalobos et al. [8].

To the best of our knowledge, no previous measured data are available for overall PAR interception in SHD olive orchards. Connor et al. [24], improving on a previous version of their model [25,26], reported model-estimated values of intercepted PAR for different SHD olive orchards, with different row spacing and orientation. For designs similar to ours (i.e., similar row spacing and orientation), they estimated values of intercepted PAR of 0.62–0.64 in the fruit growing period, not far from the 0.57 reported here for the March–September period. The small difference could be due to the slight differences in measuring periods (i.e., later periods, as in Connor et al. [24], result in greater interception, due to canopy growth, see Figure 2d), orchard slope and aspect, canopy characteristics (e.g., height and density), or model calibration.

While overall PAR transmission and interception was quite similar for the two systems (HD and SHD), the spatial pattern of transmitted PAR across the rows, was very different (Figure 2a,b). The HD system had a much more uniform distribution of transmitted PAR along the transect, which ranged from 0.34 to 0.43 (average of the two alley centers) and the variation was not significant, while in the SHD system the variation was from 0.21 below the tree row, to 0.59 in the center of the alley (average of the two alley centers) and was highly significant ($p < 0.001$, Figure 2). This is due to the greater discontinuity of canopy cover in the SHD system, given by the narrow and dense canopy. In a tree row oriented roughly N–S, as in this experiment, the center of the alley is not shaded when incident PAR is highest during the day, resulting in high PAR transmittance. The position under the tree row, instead, is almost always shaded, especially at mid-day when, although PAR is highest, direct radiation must cross the longest paths through the canopy. The more uniform canopy cover and the taller canopy of the HD system (Figure 1) reduce this phenomenon, resulting in more uniform PAR transmission on the orchard floor (Figure 2a,b). Therefore, despite quite similar overall PAR interception and transmission, the two systems utilized the radiation very differently. The SHD system "wasted" more radiation in the center of the alley where there is no canopy [31,32], resulting in greater PAR transmission in this area. This greater PAR transmission was mostly compensated for by greater interception in the area of the orchard where the canopy is concentrated, thus resulting in similar overall transmission across the orchard floor, but much denser shade under the trees. To better understand this, we can use Jackson and Palmer's [33] simplified approach to calculate light transmission (Equations (1) and (2)), in which they distinguished light transmitted directly to the orchard floor between the rows and light transmitted through the tree canopies.

$$T = Tf + Tc \tag{1}$$

and

$$Tc = (1 - Tf)\, e^{-KL'} \tag{2}$$

where T = total light transmission; Tf = fractional transmission between tree canopies; Tc = fractional transmission of light passing through the canopies; K = extinction coefficient (light intercepted per unit leaf area); L' = adjusted leaf area index (LAI/l−Tf) expressing leaf area on the basis of mean ground area shaded by the tree. At equal T, the SHD system has higher Tf (i.e., wasted light) and lower Tc, due to greater L'.

This implies greater self-shading and lower PAR levels at the bottom of the canopy in SHD systems compared to systems with a more horizontally uniform canopy. This concept is illustrated in Figure 6 where a theoretical canopy with perfectly uniform horizontal canopy cover is illustrated in the top scheme. In this theoretical canopy, the PAR transmitted on the orchard floor is uniformly distributed in space and the overall PAR transmitted is assumed to be 0.43 (i.e., the same found for the SHD system in this experiment). In this situation, self-shading is uniform and bottom-canopy leaves are uniformly illuminated across the orchard; T = 0.43, Tf = 0, Tc = 0.43 and L' is the minimum possible for that light transmittance level (see Equation (1)) implying the lowest possible level of self-shading. The lower scheme in Figure 6 shows instead the actual canopy of the SHD system in this experiment, with the values of PAR transmission under the trees (PAR = 0.21 under the tree row; PAR = 0.59 in the alley center) as found in this experiment. While the overall PAR transmittance (and therefore PAR interception), is the same in both schemes (overall transmitted PAR = 0.43), the irradiance below the tree row is less than half (i.e., 0.21 vs. 0.59). Using Equation 1, while T remains 0.43, Tf >> 0, thus Tc << 0.43, implying that L' is much higher than for HD. This implies that bottom-canopy leaves are exposed to much lower PAR levels than in the top scheme, even though overall PAR transmission is the same. The much greater spatial uniformity of measured transmitted PAR values in the HD orchard in this study, compared to the SHD system (Figure 2a,b), suggests that the HD orchard here considered was close to the theoretical scenario (i.e., uniform interception) shown in the top scheme of Figure 6, resulting in lower self-shading than in the more heterogeneous canopy cover of the SHD system.

Greater self-shading in SHD systems might results in below-optimal or even insufficient irradiance for fruit production. In fact, irradiance decreases with canopy depth and intense shade can depress both flowering and fruiting processes [5,6,19–21]. In olive, with irradiance below 40% of incident PAR, fruit become the priority sink for photosynthate at the detriment of vegetative growth, and at the same time the growth rate of the fruit and their oil concentration is reduced [23–25]. A radiation level of at least 20–30% of incident PAR is required to allow the physiological activation of the sequential steps, including sprouting, flower induction, flowering, fruit formation and filling, that lead to the reproductive sequence of the plant [26]. Below 20% of PAR there is a significant increase in fruit abscission [18], while below 10% PAR, leaf senescence occurs [22]. These concepts were discussed also by Casanova-Gascon et al. [34] who compared SHD vs. open center almond trees. Based on their transmitted light measurements, these authors speculated that bottom-canopy leaves in the SHD system had better illumination. However, this was due to large inter-row spacing, resulting in reduced canopy PAR interception (54%) compared to 74% in the open center system. Likely, at equal PAR interception, bottom-canopy leaves would be less well lit also in SHD almonds.

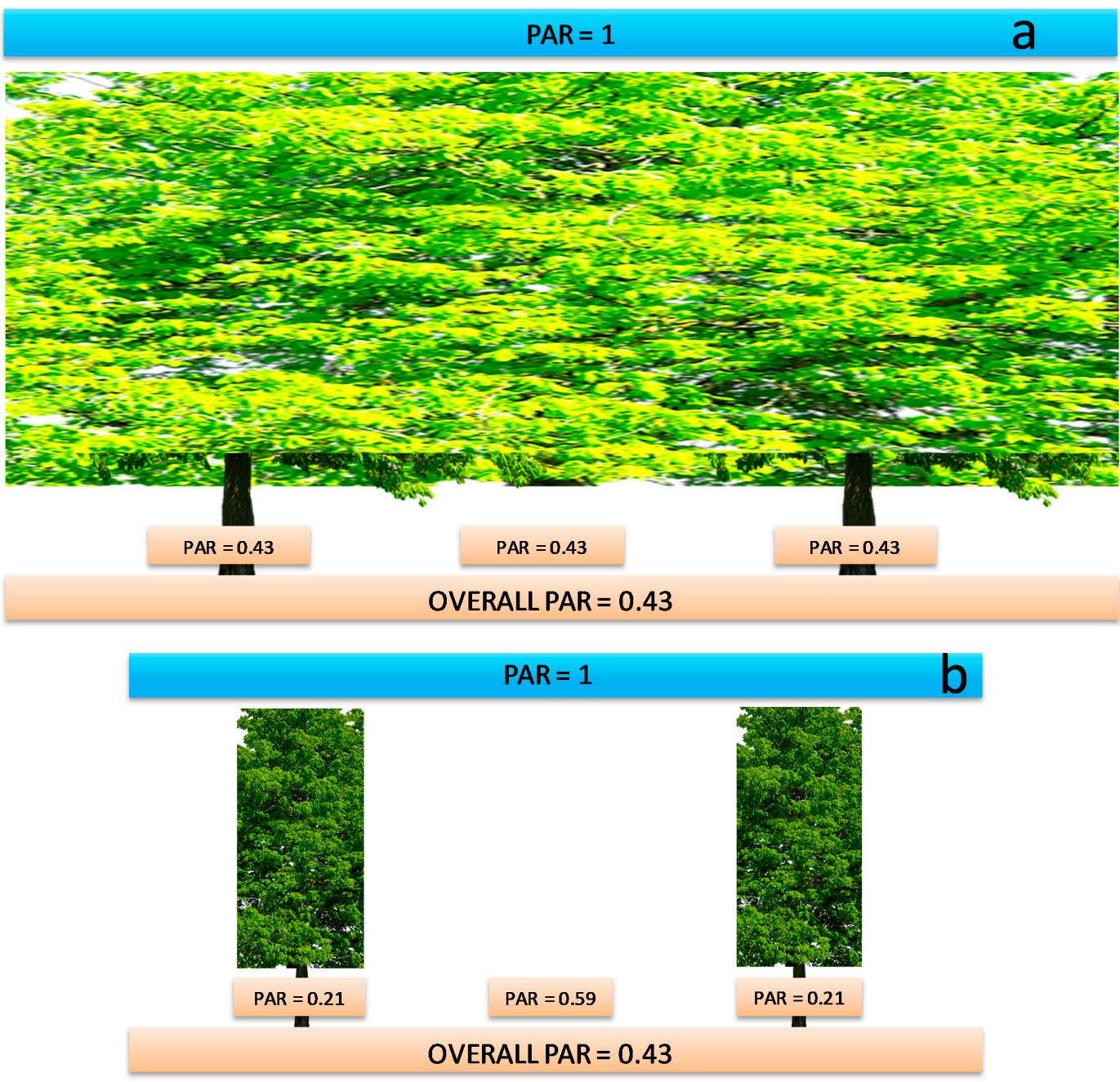

**Figure 6.** (**a**) Theoretical canopy with perfectly uniform horizontal canopy cover. In this canopy, the PAR transmitted on the orchard floor is uniformly distributed in space and is assumed to be 0.43 (i.e., the same found for the SHD system in this experiment). (**b**) Canopy of the SHD system in this experiment, with actual values of PAR transmission under the trees (PAR = 0.21 under the tree row; PAR = 0.59 in the alley centers) as measured in this experiment. In this situation, while the overall PAR transmittance (and therefore PAR interception) is the same as in the top figure (overall transmitted PAR = 0.43), the irradiance below the tree row is less than half (i.e., 0.21 vs. 0.43).

### 4.2. Implications of Different Frequency Distributions of Transmitted PAR Values

Compared to the SHD system, the more homogeneous spatial distribution of transmitted PAR in the HD system (Figure 2) resulted in greater uniformity, among the different positions, also in terms of frequency distribution patterns (Figure 4). Additionally, the positions under the tree row in the SHD system experienced lower frequencies of intermediate and high PAR levels, and higher frequencies of low PAR levels (i.e., PAR < 200 $\mu$mol m$^{-2}$ s$^{-1}$) (Figure 5b). The frequency distribution of PAR values measured below the tree row in the

SHD system is very similar to that found by Larbi et al. [29] for the PAR incident inside the bottom of a SHD canopy, with highest frequencies for very low PAR values. Therefore, the bottom side of SHD canopies appear to be exposed not only to lower daily and seasonal PAR, but also to higher frequencies of very low PAR levels, compared to HD systems. This is likely to result in lower radiation use efficiency (RUE). In fact, net assimilation is more efficient (i.e., greater RUE) at intermediate PAR levels than at both very high (due to saturation) and very low (due to leaf respiration) PAR levels [35,36]. More time spent at intermediate irradiance results in greater radiation use efficiency than at extreme levels [37]. Leaf acclimation to different irradiance levels may reduce variations in RUE within canopies, by distributing leaf nitrogen (and thus photosynthetic properties) in relation to the light gradient, resulting in optimized daily canopy photosynthesis and a linear relationship between daily photosynthesis and incident PAR [36,38–41]. Leaf acclimation to the irradiance gradient within the canopy occurs also in olive [29]. However, there is a limit to leaf acclimation at low PAR, and when too much time is spent at PAR level approaching the light compensation point (LCP), photosynthesis and RUE both decrease, becoming negative below the LCP. LCP in olive is about 50 $\mu$mol m$^{-2}$ s$^{-1}$ [30]. Both our data under the canopy and the data of Larbi et al. [29], measured at the bottom of a SHD canopy, show that the most frequent PAR values are those close to this LCP. Therefore, it is likely that these least illuminated portions of the canopy have lower RUE than better illuminated portions.

Reduced RUE in bottom-canopy leaves might contribute to explain why whole-canopy RUE (both when expressed on biomass, glucose equivalent or oil basis) decreases at increasing canopy PAR interception [8], and why Connor et al. [24] found lower oil-based RUE (i.e., 0.12 g oil/MJ PAR) in SHD systems than Villalobos et al. [8], did for extensive and HD systems (range: 0.25–0.14 g oil/MJ PAR), even when compared at the same interception level. The explanation could be that increasing interception results in greater self-shading and thus decreasing RUE, at least in the most shaded parts of the canopy, due to increasing frequency of very low PAR levels, near or below the LCP, at which RUE is lower or negative. At the same canopy PAR interception level, greater self-shading in SHD than in HD systems would imply lower RUE. Additionally, lower fruit and oil RUE at increasing shade could be explained by the fact that PAR levels are too low, in the most shaded parts of the canopy, to allow proper formation of fruit and oil, thus reducing fruit RUE down to nil at PAR values below the threshold for fruit formation. Again, at equal overall interception, greater self-shading in the lower part of the SHD canopy would reduce fruit and oil RUE more than in HD systems.

No data report variability in RUE within the olive canopy, to support the hypothesis that RUE decreases at increasing shade. However, Trentacoste et al. [28], report data on fruit density, fruit weight and oil content for individual SHD canopy layers for which they also report intercepted PAR. Fruit density decreased with canopy depth, and so did oil content and fruit weight, thus fruit and oil yield decreased even more dramatically with canopy depth. Recalculating their data to obtain both fruit and oil weight per mole of PAR intercepted, we calculated the fruit and oil RUE for each layer (Figure 7).

Both fruit and oil RUE were greater in upper canopy layers, except for the top layer where pruning stimulates less dense and more vegetative regrowth [25]. This layer can be viewed like the elongation shoots in apple, intercepting light that is not used for fruit production, resulting in a partial decoupling of total light interception and yield (5–6). However, while elongation shoots are necessary in apple for regenerating new spurs and thus for future production, the top growth in SHD olive systems will be pruned before producing fruit, thus representing unfruitful light interception. Aside from the top layer, both fruit and oil RUE decrease with canopy depth. Similar results are obtained (data not shown) if data on yield per layer in several olive orchards from Trentacoste et al. [42,43] are combined with PAR data from the same experiments, published by Connor et al. [24]. Even though the different methodologies and periods considered do not allow comparison of the recalculated RUE values across studies, the results support the hypothesis that increasing shade at increasing canopy depth may reduce fruit and oil RUE. It may be

argued that calculating RUE separately for canopy layers may be biased by assimilate translocation between layers. However, this translocation is much slower in olive than in other species [44] and a large impact of translocation between canopy layers can be excluded. Source-sink experiments show that in olive the shoot "can be considered an independent unit because it is influenced very little or not at all by conditions that occur in adjacent parts" and "its growth is influenced by the source-sink conditions within itself" [45]. Most assimilates for olive fruit development are supplied by the leaves on the same shoot where the fruit is attached [45–47]. Finally, if significant amounts of assimilates were translocated across canopy layers, this would increase RUE in the most shaded layers, where production would be supported by imported assimilates in addition to those produced with the low local light. Vice versa, RUE would be reduced in the most illuminated parts, depleted of their own assimilates. Therefore, if the RUE calculations in Figure 7 were biased by ignoring translocation of assimilates between layers, the actual RUE would decrease even more dramatically with canopy depth.

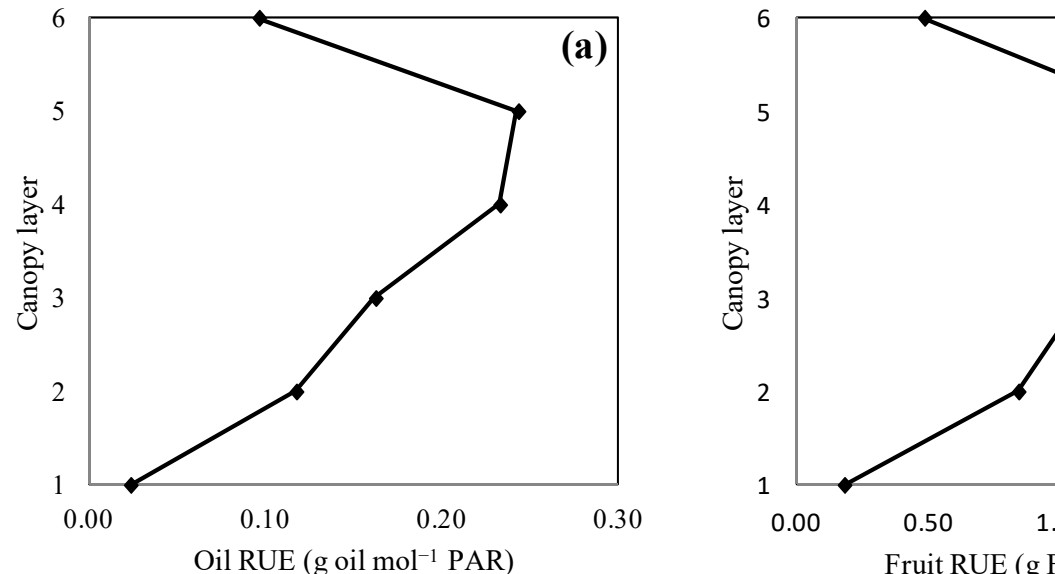 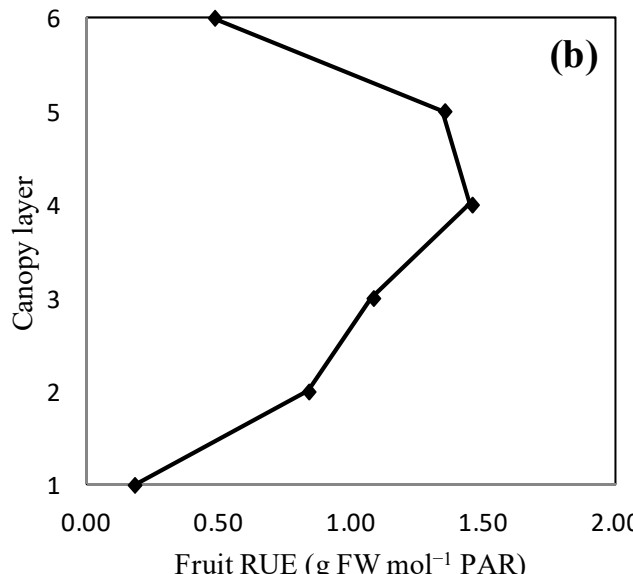

**Figure 7.** Variation of (**a**) Oil radiation use efficiency (RUE) and (**b**) Fruit RUE in the canopy profile, from the top (i.e., layer 6) to the bottom (layer 1) of the canopy. Both Oil and Fruit RUE were calculated from data on fruit density, fruit weight and oil content, and data on PAR interception over the fruiting period for individual layers, reported in Trentacoste et al. [28].

Decreasing RUE with increasing shade within the canopy implies that, at equal overall PAR interception, greater self-shading in SHD systems may result in lower RUE than in HD systems. Unfortunately, there are virtually no studies directly comparing HD and SHD systems in terms of fruit distribution within the canopy. The only study doing so [17] reports a more uniform distribution of fruit among canopy layers in a HD (i.e., 7 m by 3.5 m: 408 tree ha$^{-1}$), where fruit was equally distributed in the top three layers and some fruit was present in the fourth and lowest layer, compared to a SHD (i.e., 3.5 m by 1.5 m: 1904 tree ha$^{-1}$), where most of the fruit was concentrated in the top two layers and virtually no fruit was present in the fourth (i.e., lowest) layer. However, these results were for a single year and need to be confirmed. Further studies are also needed to confirm whether RUE (including biomass RUE, in addition to fruit and oil RUE), changes with canopy depth and PAR levels, as suggested by the recalculations in Figure 7.

## 5. Conclusions

Most previous studies on light interception and productivity in olive orchards focused on the whole-canopy and whole-orchard level. The main novelty of this work is the suggestion that knowing the overall radiation interception of an orchard is not sufficient to

optimize orchard design, as argued for other fruit orchards. While the two olive systems in our study (i.e., high density, HD, and super high density or hedgerow, SHD) had similar overall PAR interception (0.62 vs. 0.57, respectively), the SHD system had greater variability, both in space (along the transect across the rows) and in time (frequency distribution), of the PAR values transmitted at different positions on the ground, compared to HD system (Figure 2). The greater variability in space in SHD systems results in high PAR transmittance in the alley (i.e., "wasting" potentially useful PAR) and lower transmittance under the canopy, implying greater canopy self-shading, with possible negative effects on fruit production and quality in the lower parts of the canopy if the orchards is not well designed. Additionally, the greater variability in time might result in longer exposure of lower-canopy leaves to very low instantaneous PAR levels, thus reducing not only total irradiance, but possibly also its use efficiency. This might explain, at least partly, why the radiation use efficiency (RUE) reported for SHD systems is lower than that reported for HD systems and, more generally, why RUE decreases at increasing interception (which implies increasing self-shading).

Bottom-canopy illumination in SHD systems can be improved by improving their design, for example with wider inter-row spacing or shorter/thinner canopies. However, while this might improve RUE, it could reduce overall PAR interception, possibly reducing yield per unit of land area. Therefore, the guiding principle we can learn from the present study is that optimizing orchard design entails increasing light interception without decreasing RUE (due to excessive self-shading). In other words, the design should seek the best trade-off between maximizing PAR interception and maximizing its use efficiency. To achieve this, further studies are needed to better understand RUE variation within the canopy and across different orchard architectural designs.

**Author Contributions:** Conceptualization, A.R., D.M. (Dario Mantovani) and F.F.; Data curation, A.R., D.M. (Damiano Marchionni) and D.M. (Dario Mantovani); Funding acquisition, A.R. and L.P.; Methodology, A.R. and D.M. (Dario Mantovani); Supervision, A.R.; Visualization, D.M. (Damiano Marchionni); Writing—original draft, D.M. (Dario Mantovani) and D.M. (Damiano Marchionni); Writing—review & editing, A.R., L.P. and F.F. All authors have read and agreed to the published version of the manuscript.

**Funding:** This work was supported by the European Commission within the 7th Framework Program (Grant 613520, Project AGFORWARD), and within the Horizon 2020 Research and Innovation Programme (Grant agreement No. 776467 project MED-GOLD), and by the Italian Ministry of Agricultural, Food and Forestry Policies (Project MOLTI, DM 13938).

**Institutional Review Board Statement:** Not applicable.

**Acknowledgments:** We thank Darcy Gordon, for English editing.

**Conflicts of Interest:** The authors declare no conflict of interest.

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
