# Peer review of "Intercepted Photosynthetically Active Radiation (PAR) and Spatial and Temporal Distribution of Transmitted PAR under High-Density and Super High-Density Olive Orchards"

_agriculture, doi:10.3390/agriculture11040351_

Round 1

Reviewer 1 Report

Dear authors, 
Needless to say, your manuscripts always represent very useful information and innovation for an olive growing "of the future". My greetings!! These type of manuscripts are very important for a better understanding of the mechanisms underlying the physiology of production systems. I read carefully your paper, and I add some comment/suggestions in pdf attached. 
The introduction well cover the recent bibliography, the methods are well described (with some minor revisions) and the results are good presented and well discussed. 

I just suggest a good revision of the english: Even if I'm not an english mother togue, I think an english revision is necessary. 

Best regards

Author Response

Dear authors, 
Needless to say, your manuscripts always represent very useful information and innovation for an olive growing "of the future". My greetings!! These type of manuscripts are very important for a better understanding of the mechanisms underlying the physiology of production systems. I read carefully your paper, and I add some comment/suggestions in pdf attached. 
The introduction well cover the recent bibliography, the methods are well described (with some minor revisions) and the results are good presented and well discussed. 

I just suggest a good revision of the english: Even if I'm not an english mother togue, I think an english revision is necessary. 

ANSWER

Thank you for your very positive comments. We addressed all points raised as comments in the manuscript file.

In particular:

For all the small corrections suggested on the language, we either corrected as suggested or asked our language editor (native speaker colleague with PhD in Pomology from UCDavis, as acknowledged in the manuscript). We had Dr. Gordon check the whole paper.

We changed the title and keywords as suggested.

We changed the text in the objectives (end of introduction) as suggested (lines 93-96) (with minor changes for correct English).

Yes, aspect is correct.

We added pictures of the two orchards, as requested, in addition to schematic representation.

We corrected “mol m-2” as “mol m-2 season-1”. Thanks for pointing out the problem: we indicated mol m-2 assuming it was clear that it was per season, as we are talking about “seasonal” PAR. However, it is more appropriate to indicate “season-1” in the units.

Statistics on figures: You are right. We added statistics showing significant differences in the graphs (Fig. 2). As for figure 3, the data shown are only for individual days (no repetition), so no statistics are possible (not even standard errors). They are just examples of different possible days.

As for figure 4 and 5, even though you did not comment on them, we assume you would have the same suggestion, so we included standard errors and statistics (only SE in fig 4 because there are too many series, so the statistics are limited to the comparison of some of those series, in figure 5). Thank you again for the suggestion.

The quality of figure 6 should be the same as Fig.1, unless in the version of the MS you received contained lower quality images? Anyway, we’ll see if the printing office finds the pictures of sufficient quality. Else, we’ll try to improve them. Thank you.

Reviewer 2 Report

Mention in the abstract of 'within the canopy' seems to be the crux of this paper. The results show altering the tree density has no real affect on the light interception so in my view it is getting a reasonable light through the whole tree canopy will lead to better productivity than mucking around with the whole orchard. As to the idea of the orchard wasting light is nonsense after all light is escaping at the ends of the rows and off to the side so it does not matter if some light falls on the alley way or inter row as I know it. As it is it seems odd that virtually no light falls there anyway. 

A major concern is while there is some evidence that statistics were used but no details are given and it seems to me  a whole lot more stats are needed. For example the data in Fig. 2 c and d are very likely not significant and if all data were put through a valid stats test then I suspect there would be no differences. For the completeness of the paper is needed and then it will really emphasise the lack of any difference in light transmission. On that point, it seems more relevant to the thrust of the paper if the reverse of transmission, that is interception, would be a better way of expressing these data as what is absorbed is relevant to productivity not what is transmitted through the canopy.

I find the data in Fig 4 all of them that the data do not look persuasive - the incident PAR (PFD) looks as though all days were cloudy as there should have been high frequency of all light intensity classes but especially the mid classes 600 - 1200 μmol m-2 s-1 which seem very low so it is no surprise that under the tree it is very dark. This is especially so for the alley which has no moderate to high light intensity.

Fig. 5 suggests some more light is getting through but still the frequencies for reasonable light is very low and I cannot see much difference between alley centre and below the tree. If these data are means then where are the SEs and what statistical procedure has been used to test the null hypothesis.

It is reasonable to assume little light reaches the bottom of the canopy as red wavelengths are preferentially absorbed by upper leaves and far red light is transmitted and this goes on down through the layers until the light is fully attenuated. I reiterate, it is this issue which is far more important to improve productivity than worrying about the whole orchard.

minor comments change line 231 to .... skewed more .....

line 350 fruit is a plural word so do not add an 's' to the end

line 372 and others the word 'adaptation' is not correct as it infers genetic change the word you want for this context is 'acclimation'

line 380 that is a weird way to express light compensation it is a light value and light is always expressed as μmol (photons) m-2 s-1 nothing else

Author Response

Reviewer 2

Question

Mention in the abstract of 'within the canopy' seems to be the crux of this paper. The results show altering the tree density has no real affect on the light interception so in my view it is getting a reasonable light through the whole tree canopy will lead to better productivity than mucking around with the whole orchard. As to the idea of the orchard wasting light is nonsense after all light is escaping at the ends of the rows and off to the side so it does not matter if some light falls on the alley way or inter row as I know it. As it is it seems odd that virtually no light falls there anyway. 

ANSWER

Altering the tree density may indeed have a large effect on PAR interception, especially with individual-tree systems. For hedgerow systems, distance between rows is more important than tree density in the row. In our study it just happened that the two systems, with their tree densities and spacing, intercepted a similar amount of PAR. So the point of the paper is that, despite having similar light interception, SHD had a lot more shade under the trees (and thus within the lower canopy layers). So yes, we agree with the reviewer: for a given light interception, “getting a reasonable light through the whole tree canopy will lead to better productivity”. We agree with this statement and, in fact, this is the main point of the paper. That is: even though overall light interception might be similar between different architectural systems, this does not guarantee that illumination of bottom canopy layers is similar. In fact, at similar overall interception, SHD systems will have more shade in bottom canopy layers, and there will be more light transmission in the alley center, compared to HD systems. This reduces RUE.

It is not clear to us why wasting light in the alley centers is “nonsense” as SHD systems fail to intercept light in this area, while intercepting more light in the tree row, which results in greater self-shading in lower-canopy layers. We also do not understand how (or the meaning of) “light is escaping at the ends of the rows and off to the side”, so it is hard for us to reply to this. We did our best to improve the MS and hope the reviewer will find the overall revision, which implements all reviewers’ suggestions, satisfactory.

Question

A major concern is while there is some evidence that statistics were used but no details are given and it seems to me  a whole lot more stats are needed.

ANSWER

We agree and carried out statistics, now included in M&M and in the text and figures.

Question

For example the data in Fig. 2 c and d are very likely not significant and if all data were put through a valid stats test then I suspect there would be no differences. For the completeness of the paper is needed and then it will really emphasise the lack of any difference in light transmission.

ANSWER

Indeed, the overall interception was significantly different (stat added now to the paper), but still the difference was quite small (0.57 vs 0.63). So we agree that the emphasis of the paper is not about this small difference, but about the fact that, despite similar interception (in fact even slightly lower for SHD), there was a lot more shade under the SHD trees (and thus within the lower canopy layers). This is the point of the paper: virtually no difference in overall (all positions) interception (fig. 2 c and d), yet large differences among different positions in SHD, with much lower light below trees, implying greater shade in the lower canopy.

Question

On that point, it seems more relevant to the thrust of the paper if the reverse of transmission, that is interception, would be a better way of expressing these data as what is absorbed is relevant to productivity not what is transmitted through the canopy.

ANSWER

We agree with the concept and in fact we did report overall interception (0.57 vs 0.63). However, since the point of the paper is to show the low light levels under the SHD canopy (which suggests low PAR levels in the bottom-canopy layers), it seems clearer to show the transmitted PAR, which is lowest under the SHD canopy, rather than showing the reverse value, which would be highest under the SHD canopy.

Question

I find the data in Fig 4 all of them that the data do not look persuasive - the incident PAR (PFD) looks as though all days were cloudy as there should have been high frequency of all light intensity classes but especially the mid classes 600 - 1200 μmol m-2 s-1 which seem very low so it is no surprise that under the tree it is very dark. This is especially so for the alley which has no moderate to high light intensity.

ANSWER

We thank the reviewer for raising points of further discussion. We’ll try to explain better the situation. The frequency distribution graphs are divided in many classes. This dilutes differences, making them small within each class. Yet, if the repeated differences are summed up over several consecutive classes, this amounts to large differences. The reviewer mentioned that there are no big differences in incident PAR between cloudy and sunny days. Indeed, if we sum up the frequencies for the lowest four classes and for the highest 4 classes, we find that in cloudy days the first four classes sum up over 400 minutes vs. only 200 in the sunny days. On the contrary, the top 4 classes sum up perhaps 100 minutes vs. more than 400 minutes in the sunny days. Therefore sunny days total over 300 additional minutes (i.e. 5 hours!) at values above 1200 . That is a large difference indeed. Additionally, it must be considered that these are averages of the whole data set, which spanned from March to September, thus including months in which the light is not so high. In fact if we look at the shape of the curves for incident PAR for the two sunny days, we can see that the curves are in between the shape of the sunny day in March and the shape of the sunny day in May (shown in figure 3; when comparing this, consider the different scale between figure 3 and Figure 4).

The reason why, in the SHD alley centers, there are relatively small frequencies of moderate to high light intensities (although if we sum up all the minutes, the overall frequency is not so low) is explained in the paper: alley centers receive no shade from the trees around midday hours, when PAR is very high. They get shaded early and late during the day, when the shade is combined with low incident PAR, thus resulting in quite low transmitted PAR values. So there are relatively small frequencies of intermediate values.

Question

 Fig. 5 suggests some more light is getting through but still the frequencies for reasonable light is very low and I cannot see much difference between alley centre and below the tree. If these data are means then where are the SEs and what statistical procedure has been used to test the null hypothesis.

ANSWER

We totally agree about standard errors and statistics and we added both to the figures. We thank the reviewer for the suggestions to add them. As for “difference between alley centre and below the tree”, if we consider only the SHD, the accumulated frequencies for the lowest 4 classes totalize about 450 minutes for the alley centers vs. 680 under the tree (over 4 extra hours!), while the top four classes totalize about 270 minutes for the alley center vs. less than 20 minutes (more than 4 hours less!). Particularly noteworthy is the fact that most of the minutes in these extra 4 hours are in the highest class thus the difference in irradiance is extremely high, as already shown in figure 2. We are sorry that all of this was not as clear in the text and we now clarified all of this in the revised text by reporting these sorts of calculations as reported above. See for instance lines: 274-276 and 292-298. Thank you for raising the issue.

Question

It is reasonable to assume little light reaches the bottom of the canopy as red wavelengths are preferentially absorbed by upper leaves and far red light is transmitted and this goes on down through the layers until the light is fully attenuated. I reiterate, it is this issue which is far more important to improve productivity than worrying about the whole orchard.

ANSWER

Thank you, we totally agree, and this is indeed the point of the paper: even though at the whole-orchard level light interception maybe similar, SHD systems result in greater self-shading at the bottom of the canopy making them less efficient at equal interception.

Question

minor comments change line 231 to .... skewed more .....

ANSWER

Done, thanks.

Question

line 350 fruit is a plural word so do not add an 's' to the end

ANSWER

Done

Question

line 372 and others the word 'adaptation' is not correct as it infers genetic change the word you want for this context is 'acclimation'

ANSWER

Great, thanks! We changed it in all cases.

Question

line 380 that is a weird way to express light compensation it is a light value and light is always expressed as μmol (photons) m-2 s-1 nothing else

ANSWER

Thank you. Yes, of course. I have no idea how an additional “mmol” got left there, but it was not intended. We corrected it.

Reviewer 3 Report

It is advisable not to use abbreviations in the title. A slight change of the title to one, that does not contain the abbreviation is suggested. Abbreviations used in the abstract should be preceded by the use of the full name of the term (feature), i.e. first “ photosynthetically active radiation”, and later “PAR”.

The article is a typical modeling study. This makes it possible to trace the differences between the models. There are no objections to the cognitive value of the publication, the workshop side of the research and the methods of statistical analysis of the results. With this type of experiment, the obtained results refer to specific models, which in this case were selected arbitrarily. The trees, depending on the model, differed not only in the planting density, but also in age, height and width of the canopy.  Changing any parameter, e.g. cutting of canopy, row orientation, slope angle, will change the relationship between the models. The presented research is a valuable contribution to the development of an optimal model of the olive orchard in the future. However, as the authors themselves admit, it does not bring us much closer to establishing the relationship between PAR interception and the effectiveness of its use and requires further experience, especially if future research is to be of practical importance.

Linguistically, the work is written correctly. The style is always up to the author and it is difficult to change it. The text, however, is difficult to read due to the very long and complex sentences.

Author Response

Reviewer 3

Question

It is advisable not to use abbreviations in the title. A slight change of the title to one, that does not contain the abbreviation is suggested. Abbreviations used in the abstract should be preceded by the use of the full name of the term (feature), i.e. first “ photosynthetically active radiation”, and later “PAR”.

ANSWER

Thank you. We added “photosynthetically active radiation” in the title and abstract and put PAR between brackets at first mention. We checked to make sure that there were no other abbreviations that were not preceded by the extended definition.

Question

The article is a typical modeling study. This makes it possible to trace the differences between the models. There are no objections to the cognitive value of the publication, the workshop side of the research and the methods of statistical analysis of the results. With this type of experiment, the obtained results refer to specific models, which in this case were selected arbitrarily. The trees, depending on the model, differed not only in the planting density, but also in age, height and width of the canopy.  Changing any parameter, e.g. cutting of canopy, row orientation, slope angle, will change the relationship between the models. The presented research is a valuable contribution to the development of an optimal model of the olive orchard in the future. However, as the authors themselves admit, it does not bring us much closer to establishing the relationship between PAR interception and the effectiveness of its use and requires further experience, especially if future research is to be of practical importance.

Linguistically, the work is written correctly. The style is always up to the author and it is difficult to change it. The text, however, is difficult to read due to the very long and complex sentences.

ANSWER

Thank you for this good review. We had a native speaker help with English. Hopefully the text is not clearer.

Reviewer 4 Report

Your manuscript was mentioned the importance to measure solar irradiance of high-density olive orchards in the introduction. When I read the text of introduction, I expected to show positive effect of PAR for the super high-density olive orchard. However, the conclusion of your research was negative effect of PAR at the bottom of tree for the super high-density olive orchard. I also predicted that productivity of olive was decreased at high-density olive orchard. Consequently, the text of your manuscript was different impression between introduction and discussion. I suggest to revise the text of introduction. For example, you should introduce negative effect of productivity by tree density. Moreover, I found the important literature as following. You should quote the contents, and reflect your manuscript. Casanova-Gascón et al. (2019) was measured for almond orchard; however, results were important for your research.

Castillo-Ruiz et al. (2016) Olive Crown porosity measurement based on radiation transmittance: an assessment of pruning effect. Sensors 16: 723.

Alejandro et al. (2016) A dynamic model of potential growth of olive (Olea europaea L.) orchards. European Journal of Agronomy 74: 93-102.

Casanova-Gascón et al. (2019) Comparison of SHD and open-center training systems in almond tree orchards cv. ‘Soleta’. Agronomy 9: 874.

Also, I would like to indicate the point for the revision of your manuscript.

  1. You did not emphasize the novelty and originality of your research. I felt that explanation on the importance of your research was weak. When you revise the manuscript, you should explain the novelty and originality according the results of literature survey.

  1. In the introduction, you mentioned that “Unlike most herbaceous crops, tree crops, including olive orchards, have discontinuous canopies, making the assessment of PAR interception and transmission difficult to both model and measure”. Therefore, you did not mention the photosynthesis of herbaceous crops at first paragraph.

  1. In the introduction, you should add a reason why the measurement of PAR is important at the olive orchard.

  1. It was unclear how many sensors was used for the measurement of each transect. If you measure only one sensor for one transect, your research is no value. If you measure several sensors for one transect, you have to show error bars from Figure 3.

  1. The experimental period was from March to September. I would like to confirm that the period on the growth of olive in Spoleto was same as your experimental period. If olive can grow after October, your experimental period was insufficient.

  1. Your results was no statistical analysis. You have to conduct statistical analysis between HD and SHD.

  1. It was unclear the basis on the Figure 3. I could not understand the ratio on the sunny and cloudy days in March. Moreover, I could not understand the reason why you chose March and May in Figure 3.

  1. In the discussion, the quotation of results was not enough. When you write your results in the discussion, you have to quote the number of figures.

  1. In Figure 7, you should show the data for HD and SHD treatments.

  1. You should reconsider the meaning of Oil RUE, Fruit RUE and biomass RUE. The abbreviation of RUE was “radiation use efficiency”; however, oil, fruit and biomass were not concerned directly with radiation. It is better that the meaning of Oil RUE is Oil Productivity.

Author Response

 Reviewer 4

Question

Your manuscript was mentioned the importance to measure solar irradiance of high-density olive orchards in the introduction. When I read the text of introduction, I expected to show positive effect of PAR for the super high-density olive orchard. However, the conclusion of your research was negative effect of PAR at the bottom of tree for the super high-density olive orchard. I also predicted that productivity of olive was decreased at high-density olive orchard. Consequently, the text of your manuscript was different impression between introduction and discussion. I suggest to revise the text of introduction. For example, you should introduce negative effect of productivity by tree density. Moreover, I found the important literature as following. You should quote the contents, and reflect your manuscript. Casanova-Gascón et al. (2019) was measured for almond orchard; however, results were important for your research.

Castillo-Ruiz et al. (2016) Olive Crown porosity measurement based on radiation transmittance: an assessment of pruning effect. Sensors 16: 723.

Alejandro et al. (2016) A dynamic model of potential growth of olive (Olea europaea L.) orchards. European Journal of Agronomy 74: 93-102.

Casanova-Gascón et al. (2019) Comparison of SHD and open-center training systems in almond tree orchards cv. ‘Soleta’. Agronomy 9: 874.

ANSWER

Thank you for your comment. We studied the literature suggested and cited the almond paper in our manuscript (lines 388-394). We did not find that the other two papers reported additional information that was relevant enough to cite here. But both papers were very interesting reads, so thank you for mentioning them.

As for introducing the negative effects of tree densities on productivity, the introduction address it with the following text (lines 66-75):

“In fact, while the increase in tree density leads to greater light interception and increased productivity in the early stages of the orchard [17], mutual (and self) shading increases as the plants grow [17,18]. Too much shade can depress both flowering and fruiting processes in olive [19–21] and a drastic reduction in PAR (i.e. below 10% of the incident PAR) increases olive leaf senescence [22]. Even when fruiting occurs, yield and oil concentration decrease with irradiance below 40–60 % of incident PAR [18,23–25]. It is therefore important that the architectural design of SHD olive orchards allows irradiance levels above certain thresholds also at the base of the canopy [26]. This can be achieved by designing the orchard with adequate canopy height, slope, width and alley width [14,26].”

Question

Also, I would like to indicate the point for the revision of your manuscript.

  1. You did not emphasize the novelty and originality of your research. I felt that explanation on the importance of your research was weak. When you revise the manuscript, you should explain the novelty and originality according the results of literature survey.
  2. ANSWER

 Thank you. We are cautious about overemphasizing the novelties, because other referees, particularly referee 3, suggested that the results need future confirmation. However, we do appreciate your suggestion and added the following additional statements in the Conclusion:

“Most previous studies on light interception and productivity in olive orchards focused on the whole-canopy and whole-orchard level. The main novelty of this work is the suggestion that knowing the overall radiation interception of an orchard is not sufficient to optimize orchard design.”

  1. In the introduction, you mentioned that “Unlike most herbaceous crops, tree crops, including olive orchards, have discontinuous canopies, making the assessment of PAR interception and transmission difficult to both model and measure”. Therefore, you did not mention the photosynthesis of herbaceous crops at first paragraph.

Herbaceous crops are mentioned twice (line 39 and line 46) before the point indicated by the reviewer. So hopefully this is ok.

  1. In the introduction, you should add a reason why the measurement of PAR is important at the olive orchard.

 Thank you. The introduction largely covers the importance of measuring PAR in general, for all crops, including tree crops, and for olive in particular (line 40, 49, 53, 57, 58) and then even more specifically for SHD systems (line 60, 62, 71, 74, 81, 89). If there is something more specific that we missed, please let us know.

  1. It was unclear how many sensors was used for the measurement of each transect. If you measure only one sensor for one transect, your research is no value. If you measure several sensors for one transect, you have to show error bars from Figure 3.

 Thank you. Line 126-7 reported that “In each orchard (i.e. HD and SHD) the PAR transmitted below the trees was measured along four transects (i.e. four replications) across tree rows”. So each position along the transect was repeated 4 times for each measuring session. Additionally, measurements were carried out for two days per month, from March to September. Therefore, overall, transect measurements were carried out for many days in many positions in the orchard, for each distance along the transect. We added more information in the figures (standard errors and number of data (n)) for each represented point. Hopefully this clarifies the issue. Thank you.

  1. The experimental period was from March to September. I would like to confirm that the period on the growth of olive in Spoleto was same as your experimental period. If olive can grow after October, your experimental period was insufficient.

Our growing season usually extends to October, though growth is minimal then. However, we believe that measurements taken over 7 of the 8 months of growth are highly representative, especially considering that most of the growth happens in May and June. Most papers dealing with PAR interception in tree crops report measurements taken perhaps only during few, usually clear, days, perhaps once or twice in the season. And often only at midday or at 2-3 times during the day. This does not invalidate their results. We measured every minute from dawn to dusk, for two days per month, for 7 months. This is much more extended than most papers we know on the subject. We believe our data can be considered sufficiently representative.

  1. Your results was no statistical analysis. You have to conduct statistical analysis between HD and SHD.

 We agree. We added it to the text (both in M&M, in the results and in the graphs).

  1. It was unclear the basis on the Figure 3. I could not understand the ratio on the sunny and cloudy days in March. Moreover, I could not understand the reason why you chose March and May in Figure 3.

Good point. It was indeed not clear. We now modified the text to explain all of this. It should be clearer now (lines 205-214, but text reported also below). Thank you again for pointing out this lack of clarity.

The class frequency distributions of the incident PAR values measured above the canopy differed substantially with season and between clear and overcast days. To exemplify this, figure 3 shows data from three sample days: two in March (one clear and one overcast) and one in May (clear). March was chosen as an example, because it was the month with the shortest (i.e. darkest) days in the dataset, and we also happen to have a clear day and a cloudy day in the same month, so we could compare them. The clear day in May was chosen to compare a clear day in a period of long days (May) with a clear day in a period of short (dark) days (March). We chose May instead of June, when days are longest, because we happened to not have perfectly clear days in June, so May had the longest clear day in the dataset.

  1. In the discussion, the quotation of results was not enough. When you write your results in the discussion, you have to quote the number of figures.

 We agree. Some reviewers believe that in the discussion one should not recall the figures because they have been explained in the result section. However, we do agree with the reviewer, so we modified the text carefully citing all figures when discussing the relative results.

  1. In Figure 7, you should show the data for HD and SHD treatments.

Figure 7 is not from our data. It reports further elaboration of data from Trentacoste et al. [28], where only SHD systems were considered. No other data is available for other olive systems.

  1. You should reconsider the meaning of Oil RUE, Fruit RUE and biomass RUE. The abbreviation of RUE was “radiation use efficiency”; however, oil, fruit and biomass were not concerned directly with radiation. It is better that the meaning of Oil RUE is Oil Productivity.

Thank you. These terms and concepts are from the literature we are discussing in this part of the discussion. We are merely reporting pervious abbreviations and concepts as published in the literature. We agree that Oil RUE might not be an ideal term, but it is what has been used in the literature (including in one of the papers suggested by the reviewer). Changing the name would be confusing, preventing the reader from understanding what we are referring to. Additionally, “productivity” does not carry the same meaning then “efficiency”. Productivity can be high with low efficiency and vice versa. In this part of the discussion we focus on the efficiency of radiation use (i.e. Oil RUE), not on oil productivity.

Round 2

Reviewer 4 Report

Your manuscript was revised according to my suggestion. On the points of no-revision, I agreed your opinion. Therefore, your manuscript was the progress of acceptance. However, I had a request before acceptance. You should conduct statistical analysis on the results of Figure 4. You already conducted statistical analysis between HD and SHD. However, I also think that the data of frequency should conduct statistical analysis (single ANOVA) among five positions (2m, 1.5m, 1m, 0.5m and Tree) and two climates (sunny and cloudy). It is better to create a new table. Moreover, I have another idea that you conduct three-way ANOVA among positions, climates, treatments (HD and SHD) and their interactions. In this case, statistical analysis written in Figures 2 and 5 can omit. Please select on the method of statistical analysis. Of course, you have to add the description on the results of statistical analysis.

Author Response

The reviewer asked for additional statistics. Here is the reviewer’s text:

Your manuscript was revised according to my suggestion. On the points of no-revision, I agreed your opinion. Therefore, your manuscript was the progress of acceptance. However, I had a request before acceptance. You should conduct statistical analysis on the results of Figure 4. You already conducted statistical analysis between HD and SHD. However, I also think that the data of frequency should conduct statistical analysis (single ANOVA) among five positions (2m, 1.5m, 1m, 0.5m and Tree) and two climates (sunny and cloudy). It is better to create a new table. Moreover, I have another idea that you conduct three-way ANOVA among positions, climates, treatments (HD and SHD) and their interactions. In this case, statistical analysis written in Figures 2 and 5 can omit. Please select on the method of statistical analysis. Of course, you have to add the description on the results of statistical analysis.

We carried out the requested statistics for Figure 4, and added a table reporting the results, exactly as requested. The MS shows these further changes highlighted with the track change option in word.

We thank the reviewer and the editor for the additional suggestions and hope the MS is now acceptable for publication.

For all authors

Adolfo Rosati